# Revisiting Backdoor Attacks on Time Series Classification in the Frequency Domain

Submission Id: 265

## Abstract

Time series classification (TSC) is a cornerstone of modern web applications, powering tasks such as financial data analysis, network traffic monitoring, and user behavior analysis. In recent years, deep neural networks (DNNs) have greatly enhanced the performance of TSC models in these critical domains. However, DNNs are vulnerable to backdoor attacks, where attackers can covertly implant triggers into models to induce malicious outcomes. Existing backdoor attacks targeting DNN-based TSC models remain elementary. In particular, early methods borrow trigger designs from computer vision, which are ineffective for time series data. More recent approaches utilize generative models for trigger generation, but at the cost of significant computational complexity.

In this work, we analyze the limitations of existing attacks and introduce an enhanced method, *FreqBack*. Drawing inspiration from the fact that DNN models inherently capture frequency domain features in time series data, we identify that improper perturbations in the frequency domain are the root cause of ineffective attacks. To address this, we propose to generate triggers both effectively and efficiently, guided by frequency analysis. FreqBack exhibits substantial performance across five models and eight datasets, achieving an impressive attack success rate of over 90%, while maintaining less than a 3% drop in model accuracy on clean data.

## CCS Concepts

• **Mathematics of computing** → **Time series analysis**; • **Computing methodologies** → **Neural networks**; • **Security and privacy** → **Domain-specific security and privacy architectures**.

## Keywords

Time Series Classification, Backdoor Attack, Frequency Domain Analysis

**ACM Reference Format:**
Anonymous Author(s). 2018. Revisiting Backdoor Attacks on Time Series Classification in the Frequency Domain. In *Proceedings of the ACM Web Conference 2025 (WWW '25)*. ACM, New York, NY, USA, 16 pages. https://doi.org/XXXXXXX.XXXXXXX

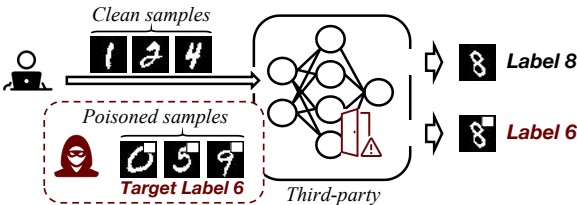

**Figure 1: The illustration of a backdoor attack.**

## 1 Introduction

Time series are ubiquitous in the landscape of web-based applications [6, 10, 13, 17, 20, 23, 28, 67], where the ability to interpret sequential data is crucial to improve user experiences and optimize service delivery. As web applications evolve, vast amounts of time series data are generated in various domains, including financial data analysis [16, 59, 71], industrial sensor monitoring [47, 55, 67], health care [7], and human activity recognition [31]. These applications rely on the accurate classification of time-dependent patterns to drive intelligent decision-making processes. The integration of time series classification (TSC) models into web technologies not only supports real-time analytics but also enables predictive capabilities that are vital for adapting to dynamic user needs and market trends [14, 24, 40, 43, 51].

Deep neural networks (DNN) have exhibited promising performance on TSC tasks [18, 36, 73]. They benefit from their superior feature extraction capability brought by the non-linear structures [66, 68]. In practice, instead of training their own models, an increasing number of users are choosing open-sourced pretrained deep learning models shared online. For example, daily downloads from model sharing platforms such as Huggingface have recently surpassed those of conventional software supply chains, including Pypi and NPM [25].

Despite the extraordinary performance of DNNs, their capability for learning complex patterns has also rendered them vulnerable to backdoor attacks. As a severe security threat for DNNs, backdoor attacks are usually conducted by a malicious third party who is commissioned to train a model [34] or provides pre-trained models or web services [33]. As an attacker, one may design specific triggers and add them to clean training samples to form poison samples. As illustrated in Fig. 1, the training process embeds the correlation between a trigger and its target label into a model. During inference, the attack can cause the model to generate expected predictions by activating the trigger, regardless of the original sample. In the meantime, since the model performance on normal samples is not affected, the backdoor in the model is stealthy.

Backdoor attacks have been extensively studied in recent years for computer vision (CV) and natural language processing (NLP) applications [34]. For instance, an attacker may easily bypass a

backdoored DNN-based facial recognition model with seemingly innocuous triggers such as wearing eyeglasses with special textures [41]. In contrast, backdoor attacks targeting time series models have received less attention. Existing works primarily focus on network traffic analysis [21] and audio recognition systems [39], while real-valued time series classification—a fundamental task—remains under-explored. This is notable since existing third-party models [2, 3] and training services [1] can be potentially vulnerable.

To better understand the backdoor threat to real-valued TSC models, one approach would be to adapt established attack methods from CV/NLP, e.g., attacks based on the static [8, 30] and dynamic triggers [35, 52] as shown in Fig. 2. **However, in Sec. 4, we find that existing methods yield underwhelming performance**, e.g., an attack success rate and classification accuracy of only 66.0% and 34.7%, respectively.[1] Other works either leverage genetic algorithms [15] or additional generative models [26] for trigger design, which comes at an extremely high computational cost.

In this work, we propose an effective and efficient backdoor attack targeting TSC models. Drawing inspiration from the observation that time series DNN models benefits from frequency domain features in the data [77], we begin by conducting an in-depth frequency domain analysis of the characteristics of both existing TSC models and backdoor attacks. Specifically, we estimate a frequency heatmap [57] to quantify the significance of each frequency band for a given model. Our analysis reveals that different models exhibit varying sensitivities to different frequency bands. A key novel insight we introduce is that **the suboptimal performance of current backdoor attacks stems from a mismatch between the frequency bands perturbed by the trigger and the frequency sensitivities of the victim model, as indicated by the heatmap**. To address this issue, we propose *FreqBack*. *FreqBack* is the first approach to integrate the *freq*uency heatmap into the trigger generation process for *back*door attacks on real-valued TSC models. Our contributions are summarized as follows:

- We are the first to analyze backdoor attacks on TSC models from the frequency domain perspective. We reveal that the trigger design of existing attacks misaligns with the model sensitivities indicated by the frequency heatmap.
- Leveraging deeper insights into the intrinsic mechanisms of TSC models and backdoor attacks in the frequency domain, we propose FreqBack, an effective and efficient backdoor attack guided by the frequency heatmap.
- Extensive empirical results on five models across eight datasets validate that FreqBack substantially improves the backdoor attack performance on TSC models. For instance, on all datasets, the average attack success rate is improved to over 90% with less than 3% degrade on clean classification accuracy.

## 2 Related Work

### 2.1 DNN-based Time Series Classification

Time series classification (TSC) categorizes a time series input to a predefined label. Along with the development of TSC, many models have been proposed to extract better features from a time series. In initial stages, human-crafted features were employed [37, 50].

---

[1]The results are on Synthetic dataset with CNN. For details of experiment settings, please refer to Sec. 6.1.

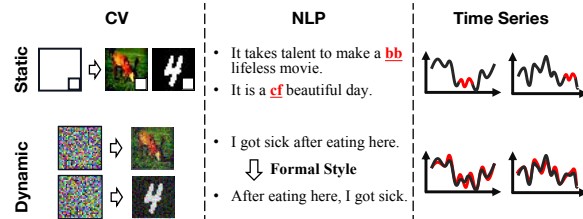

**Figure 2: Static and dynamic backdoor attacks.**

Recently, DNNs have dominated this area with their excessive modeling ability, including multi-layer perceptrons (MLP) [66], recurrent neural networks (RNN) [22, 58, 60]. Nowadays, convolutional neural networks (CNN) [18, 32] and self-attention models [70] have also proved effective for sequence modeling without suffering from gradient vanishing or explosion of RNNs.

For a dataset with $N$ labeled sequences $\mathcal{D} = \{(X^{(i)}, y^{(i)})\}_{i=1}^{N}$, we denote a classifier as $f_\theta : \mathcal{X} \to \mathcal{Y}$ with parameters $\theta$, where the samples and labels are denoted as $X^{(i)} \in \mathbb{R}^{T \times M}$ and $y^{(i)}$, respectively. We denote $X_t^{(i)} \in \mathbb{R}^M$ as the temporal input at each time-step $t \in [1, T]$, where $M$ is the input dimension. This work focuses on uni- and multi-variate TSC tasks, where $M \geq 1$.

### 2.2 Backdoor Attack

The adoption of DNNs introduces various security risks including backdoor attacks [34], where attackers manipulate the training process of the neural network to inject hidden functionalities that can be activated by specific trigger patterns. Without loss of generality, the attack objective of a backdoor attack can be summarized as:

$$\min_\theta \sum_{(X,y) \in \mathcal{D}} \ell(f_\theta(X), y) + \ell(f_\theta(\tilde{X}), \tilde{y}) \tag{1}$$

where $\ell(\cdot, \cdot)$ is the cross entropy loss, $\tilde{X}, \tilde{y}$ are poisoned samples with triggers and target labels.

Existing backdoor attacks mainly focus on CV [8, 35] and NLP [30, 52]. A few studies have been conducted on backdoors for TSC models, such as network traffic analysis [21] and speech recognition systems [39]. However, backdoor attacks on real-valued TSC models remain less explored. Existing works either require an additional generative model [26] or utilize genetic algorithms (e.g., [15]), which results in inferior efficiency in trigger generation. In this work, we present a lightweight effective attack based on frequency analysis.

### 2.3 Frequency Analysis of Time Series

DNN models are found to be able to capture frequency domain features from training data [75]. Utilizing frequency features in time series analysis has a long history as well since the waveform of time series inherently consists of frequency properties [5]. Recent works have proposed to leverage different transformation methods including DWT [64, 77], DCT [9], DFT [61, 69, 77] and FFT [42] to help with effective frequency analysis in DNN-based time series modeling. In this work, we propose to leverage frequency analysis to improve backdoor attacks for TSC models. More details about relevant works are presented in Appendix D.

# 3 Preliminaries

## 3.1 Threat Model

In this section, we introduce the threat model. In particular, we focus on the backdoor attacks induced by a malicious DNN service or training source provider.

**Attacker's Goal**: The ultimate goal of an adversary is to embed an effective and stealthy backdoor into a DNN-based TSC model. During inference, a backdoored model should output expected labels given samples with triggers. In the meantime, the model should behave normally on clean samples without triggers, so that normal users won't recognize the existence of the embedded backdoor.

**Attacker's Capability**: As a DNN service or training source provider, one can completely control the training process of a model. The trained model can be either released for public use or delivered to the client who purchased the training source. Hence, the adversary has access to the training data of the victim model, together with its architecture, e.g., RNN or CNN. During training, the attacker can manipulate the training data for backdoor implanting. To perform an attack after the model is deployed, the adversary can attach triggers to samples during inference.

## 3.2 Backdoor Attack

According to the trigger design, existing backdoor attacks can be roughly categorized into *static attacks* and *dynamic attacks*, as demonstrated in Fig. 2.

- **Static Attack**: This type of trigger has a constant pattern across all data samples. For instance, in CV tasks, a patch with a specific visual pattern is put on images as a trigger, while in NLP tasks, one may add a short term as the prefix of a sentence. To adapt such attacks to TSC, we follow previous works [26, 76] to replace a specific segment of clean samples with Gaussian noises.

- **Dynamic Attack**: This type of trigger has sample-specific patterns. To generate such targeted perturbations, one may leverage the Fast Gradient Sign Method (FGSM) [19] for images [48] or use word/sentence-level substitutions for sentences [12, 30]. With the targeted trigger, dynamic attacks can achieve better performance compared to static ones [4, 11, 48, 49]. In this work, we apply targeted FGSM [19] and PGD [45] for TSC.

# 4 Revisiting Backdoor Attacks on TSC

In this section, we conduct preliminary experiments to explore why existing backdoor attacks have less-than-satisfactory performance. Inspired by the finding that time series DNN models can capture the frequency domain features [77], we propose to explore the model sensitivity to different frequency bands in the first place. Then, we evaluate existing attacks from the frequency perspective and show the reason behind the unsatisfactory performance.

## 4.1 Preliminary Results

To evaluate the performance of existing methods, we first conduct experiments with three representative attacks, i.e., the Static attack, the Dynamic (PGD) attack, and the generative attack on TSC models, TSBA [26].[2]

---

[2]For preliminary experiments, we consider a random target label setting. For detailed experimental settings and results, please refer to Sec. 6.

As shown in Table. 1 from Sec. 6, all three attacks turn out to have not good enough performance across datasets. For instance, in terms of the attack success rate (ASR), the Static attack only achieves an average ASR of 44.8% on the UWave dataset. The PGD attack tends to be more effective, where the average ASR is 62.0%. However, the performance is still far from satisfactory. Moreover, for TSBA, the results state that the average ASR is consistently below 50% except on the Eye dataset.

On the other hand, in terms of the classification accuracy (ACC) of the clean samples, all three methods damage ACC to varying degrees on different datasets. For instance, the average ACC drops by 18.1%, 5.8%, 12.7% for the Static attack on the UWave dataset, the PGD attack on the Epilepsy dataset, and the TSBA attack on the Eye dataset.

**In brief, existing backdoor attacks on TSC models have suboptimal performance in terms of both ASR and ACC.** Below, we explore the reason behind with frequency domain analysis.

## 4.2 Frequency Sensitivity of Models

In this work, to investigate the reason why existing attacks have poor performance, we propose to leverage frequency domain analysis tools. Specifically, inspired by previous work in computer vision [72], we use a frequency heatmap to estimate the model sensitivity to different frequency bands given a set of data samples. For the design details of generating the frequency heatmap, please refer to Sec. 5.1. At present, we simply give out the generated heatmaps in the first five subplots of Fig. 3.

These plots illustrate how different model architectures are sensitive to various frequency bands. Each bar in a subplot represents the average loss increase over the whole data set when a perturbation on the corresponding frequency basis at the band is added to the sample, i.e., model sensitivity. From the plots, we can tell that the frequency domain sensitivity of models correlates highly with the model architecture. According to the definition, the left and right ends of the plot stand for the low-frequency bands, while the center part represents the high ones. Therefore, **RNN-based models (BiRNN, LSTM) tend to be more sensitive to low-frequency perturbations, while CNN and self-attention (DynamicConv) models are more sensitive to midrange-/high-frequency perturbations.**

## 4.3 Existing Attacks from the Frequency Perspective

Distinct model sensitivity in the frequency domain for different model architectures implies that one universal trigger pattern in the temporal domain may not work well for a backdoor attack. To further explore if it is such a mismatch in frequency bands of the trigger pattern and the actual model sensitivity that causes the unsatisfying attack performance, we visualize the average perturbation scale of the attacks on the BiRNN model. We show the results in the last three subplots of Fig. 3. To summarize the actual perturbation scale, absolute values are taken for visualization.

Comparing the pattern of the perturbations in the frequency domain with the model frequency heatmap of BiRNN (i.e., the second row in Fig. 3), we can find that all three attacks exhibit an inconsistent pattern with the heatmap. For instance, the Static and

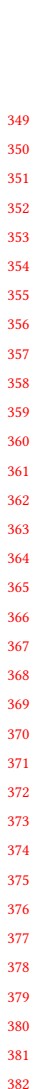
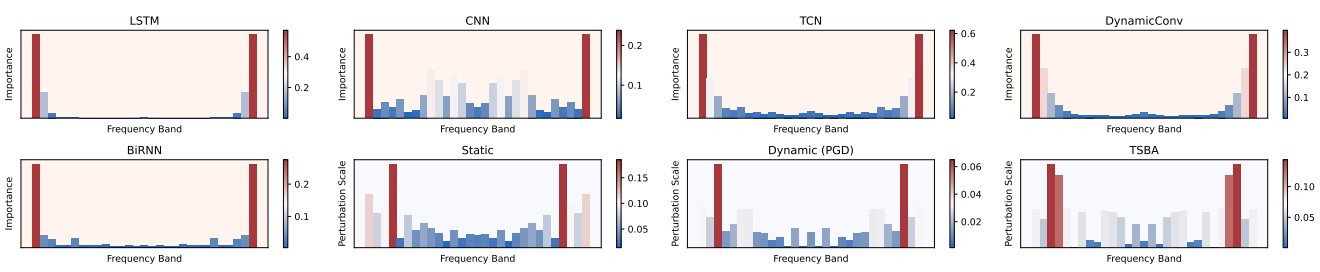

**Figure 3: The frequency domain analysis conducted on the RacketSports dataset. The first five subplots demonstrate the frequency heatmap generated for different model architectures. The last three subplots demonstrate the average perturbation scale in the frequency domain of the Static, PGD, and TSBA attacks on the BiRNN model. The values are averaged over channels.**

PGD attacks have more midrange-frequency perturbations, while the TSBA attack has more midrange/high-frequency perturbations. **All of them deviate from the sensitive low-frequency bands of BiRNN. Such a phenomenon is shared across datasets. We believe that this is the root cause of the less-than-satisfactory performance of current backdoor attack methods.** For more frequency heatmaps and perturbation scale visualizations on other datasets and models, please refer to Appendix B.5.

## 5  Our Approach

In this work, we propose to analyze and enhance backdoor attacks on TSC models from a frequency domain perspective. To this end, we first describe how to estimate the frequency heatmap for a specific model given a set of data samples in Sec. 5.1. Then, we propose our trigger generation design based on the estimated frequency heatmap in Sec. 5.2. Finally, we summarize the overall backdoor embedding procedure in Sec. 5.3.

### 5.1  Frequency Heatmap

*Discrete Fourier Transformation.* The estimation of the frequency heatmap for TSC models is based on transformation between temporal domain data and frequency domain data. In this work, we leverage the commonly used 1D discrete Fourier transform (DFT) [61, 69, 77]. Formally, for a multivariate time series $X \in \mathbb{R}^{T \times M}$, where $T$ denotes the length of the time-steps and $M$ denotes the number of channels, the 1D DFT transforms each channel of the sample from temporal domain to frequency domain, i.e., $\mathcal{F} : \mathbb{R}^T \to \mathbb{C}^T$. Similarly, the 1D inverse DFT (iDFT) transforms the frequency domain data back to the temporal domain, $\mathcal{F}^{-1} : \mathbb{C}^T \to \mathbb{R}^T$. For a univariate time series $X_{*,m}$ from $X$ at channel $m$, the detailed transformation of $\mathcal{F}$ and $\mathcal{F}^{-1}$ is represented as $\mathcal{F} : F_{t',m} = \frac{1}{T^2} \sum_{t=0}^{T} X_{t,m} \cdot e^{-i\frac{2\pi t}{T}t'}$, $\mathcal{F}^{-1} : X_{t,m} = \frac{1}{T^2} \sum_{t'=0}^{T} F_{t',m} \cdot e^{-i\frac{2\pi t'}{T}t}$.

*Frequency Heatmap Estimation.* To evaluate the frequency sensitivity of a model, we gain insight from previous work in computer vision [72] and propose to estimate a frequency heatmap. The heatmap renders how the model output is impacted by perturbations on a certain frequency band.

Practically, to estimate the model sensitivity $S_t$ to frequency band $t \in [1, \cdots, T]$, we first construct a temporal domain perturbation basis vector $U_t \in \mathbb{R}^T$, such that $\mathcal{F}(U_t)$ in the frequency domain only has up to two non-zero elements being symmetric to the center

of the sequence. [3] Next, a sample perturbed at frequency band $t$ can be achieved by,

$$\tilde{X}_t = X + \lambda \cdot U_t, \tag{2}$$

where $\lambda$ is a hyper-parameter adjusting the norm of the perturbation. Then, the sensitivity $S_t$ of model $f_\theta$ is then measured by,

$$S_t = \frac{1}{||\mathcal{D}||} \sum_{(X,y) \in \mathcal{D}} \ell(f_\theta(\tilde{X}_t), y) - \ell(f_\theta(X, y)), \tag{3}$$

where $\ell(\cdot, \cdot)$ denotes the cross entropy loss for TSC tasks, $\mathcal{D}$ denotes the set of samples for estimating the model sensitivity. Iteratively estimating $S_{t,m}, t \in [1, \cdots, T], m \in [1, \cdots, M]$ provides us with the complete frequency heatmap of a model on a set of samples. Therefore, the complexity of the heatmap estimation is $O(||\mathcal{D}||TM)$, where $||\mathcal{D}||$ denotes the number of samples. In practice, the heatmap only needs to be estimated for one time after the model completes training. Moreover, the inference of samples can be done in parallel on GPUs, which can largely shorten the time cost.

### 5.2  Trigger Generation

Based on the analysis in Sec. 4, we point out that the unsatisfying performance of current backdoor attacks lies in the mismatch of the trigger pattern and the model sensitivity in the frequency domain. To fill the gap, we propose **FreqBack**, which leverages the victim model's **freq**uency heatmap for the **back**door trigger generation.

In this work, based on the previous analysis that models extract frequency domain features, we propose to initialize and optimize the trigger $p'$ in the frequency domain. Particularly, we propose the following frequency domain objective,

$$\min_{p'} L_{\text{CE}}(\tilde{X}, \tilde{y}; \theta) + \alpha \cdot L_{\text{Freq}}(p', S; \theta) + \beta \cdot L_{\text{Regularization}}(p'), \tag{4}$$

where $\tilde{X} = X + \mathcal{F}^{-1}(p')$, $\alpha, \beta$ are the hyperparameters controlling the weight of different objective terms, and $L_{\text{CE}}$ is cross entropy loss. To instruct the generated trigger to perturb the most sensitive frequency bands of $f_\theta$, we define $L_{\text{Freq}}$ as follows,

$$L_{\text{Freq}}(p', S; \theta) = \frac{1}{M} \sum_m ||S_{*,m} - p'_{*,m}||_2^2, \tag{5}$$

where $S$ is the frequency heatmap estimated on $f_\theta$ with the clean training set. To further regularize the perturbation scale in both

---

[3]Note that we estimate the model sensitivity channel by channel in practice. For simplicity, we omit the channel index $m$ in the notations.

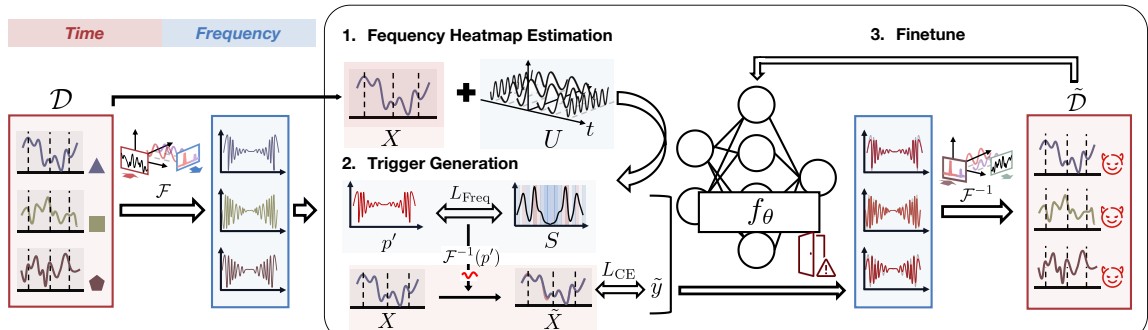

**Figure 4: The overview of the procedure of FreqBack. In each iteration: (1) Time series data are transformed to frequency domain for frequency heatmap estimation on victim model. (2) Triggers are generated under the guidance of the heatmap and other regularizations in both domains. (3) Poison data with triggers are used to finetune the victim model.**

the frequency domain and the temporal domain, we include the regularization term, $L_{\text{Regularization}}(p') = ||p'||_2^2 + ||\mathcal{F}^{-1}(p')||_2^2$. This regularization term restricts the trigger from introducing obvious perturbations to the sample in both temporal and frequency domain. With the objective in Eq. 4, FreqBack can iteratively optimize a trigger by common optimization methods, e.g., Gradient Descent.

## 5.3 Overall Procedure

Based on Eq. 4, FreqBack can generate effective trigger patterns for each sample. Then, we use these triggers to construct the poisoned dataset and train the backdoor model according to the objective in Eq. 1. In this work, to further boost the performance of the backdoor attack, we utilize an iterative backdoor implanting procedure. Fig. 4 depicts one iteration of training: for a victim model $f_\theta$, we first generate poisoned samples based on Eq. 4. Next, we finetune $f_\theta$ for a backdoored model $f_\theta'$. In the next iteration, the poisoned samples generation is based on the new model $f_\theta'$. In this way, FreqBack quickly finds the optimal trigger pattern since both the model and the trigger are optimized toward the same objective. The overall algorithm is summarized in Appendix C.

## 6 Experiments

In this section, we conduct experiments on multiple models with different architectures across various time-series datasets. Through the experiments, we answer the following research questions:
**RQ1**: Can our proposed FreqBack perform effective backdoor attacks on TSC models?
**RQ2**: How does the frequency heatmap improve the generation of the backdoor trigger?
**RQ3**: How efficient is FreqBack compared to SOTA baselines?
**RQ4**: How do potential backdoor defense strategies affect the attack performance of our FreqBack?

## 6.1 Experimental Setting

*Datasets.* In this work, we consider the following eight datasets simulating various web scenarios. **Synthetic**: a pseudo periodic synthetic dataset, simulating web traffic patterns or periodic user interactions. **Climate**: a dataset capturing green house gas concentrations, consisting of "Green Gas Observing Network Dataset"

[44] and "Atmospheric Co2 Dataset" [29], relevant for environment monitoring web services. **Stock**: a stock price dataset containing collected stock price of 564 corporations in the Nasdaq Stock Market from September 30, 2003, to December 29, 2017, with one sample per week, highlighting real-time financial data analysis, a key feature in many web applications. **ElectricDevices**: an electric device dataset that involves household devices in seven classes of screen group, dishwasher, cold group, immersion heater, kettle, oven cooker, and washing machine, reflecting smart home IoT data, which is commonly processed by web-based applications. **Epilepsy**: a movement dataset that involves healthy participants engaging in four classes of activities: walking, sawing, running, and seizures, measured using a tri-axial accelerometer, relevant to health monitoring web applications. **RacketSports**: a movement dataset designed to predict the sport and stroke being played by a player based on the statistics captured by a smartwatch, akin to sports analytics systems. **UWave**: a gesture recognition dataset comprising the X, Y, and Z coordinates of accelerometers representing eight basic gestures, akin to sports analytics systems. **Eye**: an eye state dataset capturing the open and closing states of eyes, which is obtained from a single continuous EEG measurement using the Emotiv EEG Neuroheadset, relevant to health monitoring web applications. Among them, the former four datasets are univariate, while the latter four are multivariate.

*Models.* We consider five representative TSC models. **RNN**: BiRNN [58] and LSTM[22], which update the hidden state given a time-series input at each step by the recurrent unit. They exhibit superior capacity in capturing long-term dependencies when compared to conventional RNNs. **CNN**: CNN [18] and TCN [32]. Standard CNN treats a time series as a vector and applies convolution and pooling operations to obtain classification results. TCN (Temporal Convolutional Network) views the sequence as a sentence in NLP and captures its embedding representation at each time step. CNN structures can better model the short-term patterns via convolutions compared to conventional RNNs. **Self-attention**: DynamicConv [70] introduces local convolution operations based on TCN structure and a self-attention feature pooling to better capture local timing features.

*Baseline Attacks.* We compare our method with the following attack baselines: **Static Attack**: Resembling the static patch attack

**Table 1: Experimental results of backdoor attacks in terms of classification accuracy (ACC) and attack success rate (ASR). The best ACC/ASR under the same setting are set to bold.**

| Dataset | Classifier | Clean | Static | | FGSM | | PGD | | JSMA | | TSBA | | Ours | | TSBA | | Ours | |
|---|---|---|---|---|---|---|---|---|---|---|---|---|---|---|---|---|---|---|
| | | | | | | | Random-Label | | | | | | | | | Single-Label | | |
| | | ACC | ACC | ASR | ACC | ASR | ACC | ASR | ACC | ASR | ACC | ASR | ACC | ASR | ACC | ASR | ACC | ASR |
| Synthetic | BiRNN | 85.7 | **88.7** | **100.0** | 84.7 | 75.0 | 83.3 | 90.3 | 84.3 | 83.7 | 87.0 | 51.0 | 85.0 | 98.7 | **88.7** | **100.0** | 83.0 | **100.0** |
| | LSTM | 86.0 | 76.0 | 72.3 | 82.0 | 50.7 | 84.0 | 84.0 | 83.0 | 51.3 | 85.0 | 43.7 | **85.3** | 99.7 | 86.0 | **100.0** | 84.0 | **100.0** |
| | CNN | 83.3 | 66.0 | 34.7 | **86.7** | 79.7 | 84.7 | **80.0** | 69.3 | 70.7 | 59.7 | 37.3 | 82.3 | 79.0 | 78.3 | 98.4 | **79.0** | **99.2** |
| | TCN | 78.6 | 71.3 | 99.7 | 70.7 | 82.0 | 72.3 | **100.0** | 71.7 | 89.0 | 62.0 | 34.3 | **77.7** | **100.0** | 73.6 | 99.0 | 74.7 | **100.0** |
| | DynamicConv | 85.3 | 86.0 | 48.3 | 86.0 | 90.0 | 87.0 | **100.0** | 85.7 | 83.0 | 86.0 | 42.7 | **88.0** | **100.0** | 85.3 | **100.0** | 84.7 | **100.0** |
| | Averaged | 83.8 | 77.6 | 71.0 | 82.0 | 75.5 | 82.3 | 90.9 | 78.8 | 75.5 | 75.9 | 41.8 | **83.7** | 95.5 | 82.4 | 99.5 | 81.1 | 99.8 |
| ElectricDevices | BiRNN | 74.2 | 70.8 | 97.8 | 71.9 | 35.1 | 68.6 | 79.1 | **73.5** | 53.3 | 72.0 | 15.6 | 71.2 | **99.8** | 72.9 | 99.8 | 73.0 | 99.9 |
| | LSTM | 73.1 | 68.2 | 98.5 | 70.2 | 21.2 | 65.9 | 61.7 | 69.6 | 34.4 | 69.8 | 15.7 | **71.0** | **99.8** | 71.1 | 99.3 | 71.9 | 100.0 |
| | CNN | 60.5 | 53.1 | 98.3 | 48.6 | 18.1 | 55.1 | 60.9 | 48.6 | 9.0 | 53.7 | 11.1 | **56.3** | **100.0** | 29.2 | 96.2 | 58.8 | 97.9 |
| | TCN | 71.0 | **71.4** | 98.5 | 68.9 | 44.0 | 69.3 | 86.8 | 70.4 | 79.8 | 69.6 | 15.6 | 70.4 | 97.5 | 69.3 | 98.3 | 71.4 | 100.0 |
| | DynamicConv | 64.5 | 59.8 | 39.0 | 63.5 | 47.3 | 64.2 | 82.7 | 57.3 | 68.7 | 64.4 | 13.8 | **65.3** | 96.1 | 63.1 | **99.6** | 67.2 | 98.5 |
| | Averaged | 68.7 | 64.6 | 86.4 | 64.6 | 33.1 | 64.6 | 74.2 | 63.9 | 49.0 | 65.9 | 14.4 | **66.9** | 98.6 | 61.1 | 98.6 | 68.5 | 99.3 |
| Epilepsy | BiRNN | 96.4 | 83.6 | 47.3 | 81.8 | 38.2 | 89.1 | 83.6 | 90.9 | 81.8 | 78.2 | 21.8 | **94.6** | **100.0** | 85.5 | **100.0** | 94.6 | **100.0** |
| | LSTM | 90.9 | 63.6 | 43.6 | 76.4 | 38.2 | 83.6 | 49.1 | 85.5 | 58.2 | 76.4 | 25.5 | **89.1** | 96.4 | 80.0 | 97.6 | 89.1 | **100.0** |
| | CNN | 96.4 | 87.3 | **100.0** | 61.8 | 27.3 | 92.7 | 81.8 | 70.9 | 40.0 | 65.5 | 30.9 | **94.6** | 83.6 | 76.4 | 97.6 | 94.6 | **100.0** |
| | TCN | 96.4 | 92.7 | **98.2** | 87.3 | 43.6 | 90.9 | 94.6 | **98.2** | 92.7 | 85.5 | 21.8 | 96.4 | 92.7 | 89.1 | 92.9 | 98.2 | 97.6 |
| | DynamicConv | 94.6 | 76.4 | 27.3 | **94.6** | 70.9 | 89.1 | 89.1 | 90.9 | 87.3 | 81.8 | 14.6 | 92.7 | **98.2** | 70.9 | 76.2 | 94.6 | **100.0** |
| | Averaged | 94.9 | 80.7 | 63.3 | 80.4 | 43.6 | 89.1 | 79.6 | 87.3 | 72.0 | 77.5 | 22.9 | **93.5** | 94.2 | 80.4 | 92.9 | 94.2 | 99.5 |
| UWave | BiRNN | 86.4 | 61.4 | 28.4 | 67.1 | 23.9 | 62.5 | 53.4 | 75.0 | 56.8 | **84.1** | 15.9 | 83.0 | **95.5** | 78.4 | 98.8 | 86.4 | **100.0** |
| | LSTM | 71.6 | 44.3 | 15.9 | 45.5 | 21.6 | 47.7 | 21.6 | 68.2 | 31.8 | 54.6 | 10.2 | **70.5** | 73.9 | 77.3 | 100.0 | 62.5 | 91.6 |
| | CNN | 75.0 | 58.0 | **98.9** | 75.0 | 20.4 | 75.0 | 75.0 | 70.5 | 30.7 | 53.4 | 13.6 | **77.3** | 94.3 | 48.9 | 97.6 | 73.9 | **100.0** |
| | TCN | 86.4 | 78.4 | 56.8 | 76.1 | 29.6 | 75.0 | 70.5 | 81.8 | 54.6 | 71.6 | 12.5 | **86.4** | 94.3 | 86.4 | 98.8 | 87.5 | **100.0** |
| | DynamicConv | 73.9 | 60.2 | 23.9 | 67.1 | 56.8 | 70.5 | 89.8 | 58.0 | 70.5 | 61.4 | 6.8 | **80.7** | 97.7 | 77.3 | **100.0** | 77.3 | 94.0 |
| | Averaged | 78.6 | 60.5 | 44.8 | 66.1 | 30.4 | 66.1 | 62.0 | 70.7 | 48.9 | 65.0 | 11.8 | **79.5** | 91.1 | 73.6 | 99.0 | 77.5 | 97.1 |
| Eye | BiRNN | 96.3 | 95.0 | 97.5 | **96.3** | **100.0** | 92.5 | **100.0** | 96.3 | **100.0** | 91.3 | 81.3 | 96.3 | **100.0** | 86.3 | 93.6 | 97.5 | 100.0 |
| | LSTM | 97.5 | 50.0 | 50.0 | 93.8 | 97.5 | 93.8 | 98.8 | **96.3** | **100.0** | 80.0 | 78.8 | 95.0 | 98.8 | 90.0 | 87.1 | 95.0 | 100.0 |
| | CNN | 96.3 | 73.8 | **100.0** | 58.8 | 42.5 | **96.2** | 97.5 | 78.8 | 50.0 | 78.8 | 68.8 | 95.0 | **100.0** | 90.0 | 83.9 | 96.3 | 96.8 |
| | TCN | 93.8 | **96.3** | **100.0** | 90.0 | 95.0 | **96.3** | 98.8 | 93.8 | 96.3 | 92.5 | 95.0 | 90.0 | 96.3 | 87.5 | **96.8** | 96.3 | 96.8 |
| | DynamicConv | 96.3 | 82.5 | 77.5 | **97.5** | **100.0** | 93.8 | 98.8 | 95.0 | **100.0** | 73.8 | 68.8 | 96.3 | **100.0** | 92.5 | 90.3 | 95.0 | 100.0 |
| | Averaged | 96.0 | 79.5 | 85.0 | 87.3 | 87.0 | **94.5** | 98.8 | 92.0 | 89.3 | 83.3 | 78.5 | **94.5** | 99.0 | 89.3 | 90.3 | 96.0 | 98.7 |

in CV [8], we randomly replace a segment in a sequence for each target label with Gaussian noises to construct poisoned samples. **Dynamic Attack (FGSM, PGD)**: Similarly, we adopt common adversarial attacks FGSM [19] and PGD [45] to construct per-sample dynamic triggers. **JSMA**: We adapt the JSMA [53] attack in CV as a temporal baseline of our method. JSMA first chooses important positions and then optimizes perturbations accordingly. **TSBA**: TSBA [26] is a SOTA backdoor attack for TSC models, which leverages a generative approach for crafting stealthy sample-specific backdoor trigger patterns. **TimeTrojan**: TimeTrojan-DE [15] is another SOTA attack that leverages genetic algorithms for trigger generation.

*Attack Setting.* We consider the random-label setting, which assigns a random label other than the ground truth to each sample as the target label for poisoned samples. We also consider the single-label setting where all poisoned samples share the same target label, to facilitate comparison with the TSBA. *Note that the random-label setting is more difficult.* If an attack method can achieve good performance in the random-label setting, it can surely achieve better performance for single-label.

*Evaluation Metrics.* To measure the attack performance, we employ the attack success rate (ASR) and the classification accuracy (ACC). ASR is the ratio of poisoned samples correctly predicted as target labels, while ACC is the ratio of clean samples correctly predicted as ground truth labels. A higher ASR reflects a higher attack effectiveness. A higher ACC signifies a stealthy backdoor. Both metrics are evaluated on the test set.

For other details of datasets and implementation, please refer to Appendix A.

## 6.2 Main Results

To answer **RQ1**, we conduct thorough experiments on all the datasets and attack baselines. [4] As shown in Table 1, all DNN models achieve promising results on most of the TSC tasks (the Clean column). Note that stock price prediction is naturally hard, where all models perform poorly.

As for the comparison among backdoor attack methods, we first consider the random-label setting. From the results, we can tell that Static attacks often perform poorer than Dynamic attacks. Although Dynamic attacks like FGSM and PGD perform better with per-sample optimized triggers, our FreqBack substantially outperforms these baselines. As discussed in Sec. 4, the main reason lies in whether the perturbed frequency bands by the trigger and

---

[4]Due to page limit, we defer the result of other datasets to Appendix B.1.

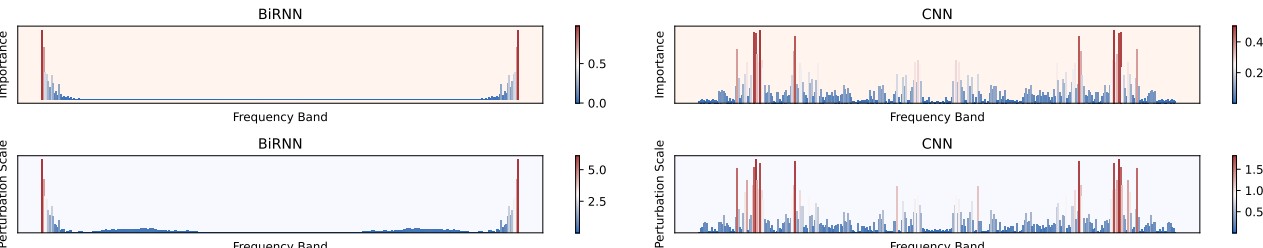

**Figure 5: The frequency heatmap (up) and perturbation scale of our method (down) for BiRNN and CNN on the UWave dataset. The values are averaged over channels.**

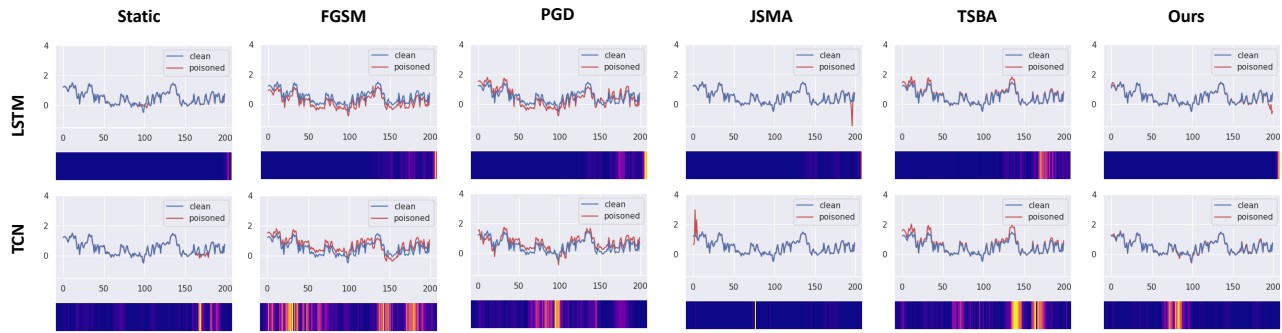

**Figure 6: The temporal domain visualization of trigger patterns and model saliency map by different attacks on the Eye dataset.**

the frequency heatmap of the backdoored model match with each other. *A less matching trigger design induced by Static or Dynamic attacks perturbs trivial positions, which contribute less to the target label, resulting in a lower ASR.*

To illustrate the effectiveness of the frequency domain trigger generation, we adapt JSMA to TSC models. From the table, we can see that JSMA has a better performance than naive dynamic attacks because it selects attack positions carefully in the temporal domain. For example, on the Epilepsy dataset, the ASR of JSMA is 72.0%, while FGSM is only 43.6%. However, *such temporal domain optimization is outperformed by the frequency domain one of FreqBack.*

As for the TSBA attack, we show that the generative model setting can not work properly in a random-label setting. We attribute this to the limited knowledge of the trigger generator about the target label, while our method does not have such limitations. Moreover, in the single-label setting, FreqBack achieved better performance across six of the eight datasets as well. Five of the ASRs reach over 99.5% and the ACCs drop by less than 3.0%. Such results state the substantial effectiveness of FreqBack.

## 6.3 Visualization of Triggers

To answer **RQ2**, we conduct visualizations in both frequency and temporal domains.

Firstly, similar to Sec. 4, we visualize the frequency heatmap and the perturbation scale of the trigger generated by FreqBack in the frequency domain in Fig. 5. In contrast to the attacks we have analyzed previously, our trigger does match the distribution of the frequency heatmap for both models.

Furthermore, to gain a more intuitive understanding of the triggers, we visualize the poisoned samples in the temporal domain in

**Table 2: Experimental results of backdoor attacks in terms of classification accuracy (ACC) and attack success rate (ASR) for both the training set and test set. The results presented are the average performance across all five classifiers.**

| Dataset | Train / Test | Static | | Dynamic (PGD) | | Ours | |
|---|---|---|---|---|---|---|---|
| | | ACC | ASR | ACC | ASR | ACC | ASR |
| Epilepsy | Train | 90.2 | 79.0 | 100.0 | 98.3 | 99.6 | 97.4 |
| | Test | 80.7 | 63.3 | 89.1 | 79.6 | **93.5** | **94.2** |
| RacketSports | Train | 94.3 | 90.9 | 98.1 | 92.4 | 98.3 | 96.4 |
| | Test | 74.8 | 71.5 | 75.4 | 73.8 | **81.6** | **93.1** |

Fig. 6. We can see that FreqBack perturbs fewer positions under smaller budgets with the guide of the frequency heatmap. This trend can be exhibited among different model architectures, which states the stealthiness of FreqBack in the temporal domain.

We also leverage the widely adopted gradients of counterfactuals method [62] to investigate the model's temporal focus on the sample. The temporal saliency map further validates that our trigger added in critical frequency bands can be recognized by models well, while the triggers of naive dynamic attacks (FGSM/PGD) are less noticed.

## 6.4 Investigation of the Backdoor Learning

To quantitatively address **RQ2**, we evaluate the average model performance on both the training and test sets for Static, Dynamic (PGD), and our FreqBack, demonstrating how frequency analysis enhances backdoor learning.

We report the ACC and ASR in Table 2. The results show a clear performance gap between the training and test sets in terms of both ACC and ASR for existing Static and Dynamic attacks, whereas our method exhibits a significantly smaller gap.

**Table 3: The running time (second) of TSBA, TimeTrojan, our method (heatmap estimation / in total) for BiRNN on the entire training set.**

| Dataset | T | Train Set Size | TSBA | TimeTrojan | Ours |
|---------|-----|------|------|-----------|---------|
| RacketSport | 30 | 242 | 19.4 | 7,690.8 | 0.2 / **3.5** |
| Epilepsy | 206 | 220 | 13.2 | 44,233.2 | 2.9 / **12.2** |
| UWave | 315 | 352 | 28.2 | 123,027.5 | 6.5 / **18.9** |

**Table 4: The backdoor performance (ACC/ASR) of TSBA and our method under Neural Cleanse defense with unlearning. '-' and '△' indicate an un-/falsely detected target label.**

| Dataset | Attack | BiRNN | LSTM | CNN | TCN | DynamicConv |
|---------|--------|-------|------|-----|-----|-------------|
| Epilepsy | TSBA | 85.5/100.0 | 78.2/54.8 | - | 56.4/0.0 | △ |
|  | Ours | - | - | - | △ | △ |
| RacketSports | TSBA | - | 80.3/60.0 | - | - | - |
|  | Ours | 80.3/100.0 | - | 73.8/93.3 | - | - |
| UWave | TSBA | - | - | - | - | - |
|  | Ours | - | 64.8/68.7 | - | △ | - |

As shown in Fig. 5, different model architectures and datasets demonstrate varying frequency preferences. We infer that current attacks fail to capture features that models can easily learn and generalize, resulting in a larger performance discrepancy between the training and test sets. **In contrast, our frequency heatmap identifies the most effective frequency bands to perturb to deviate the model output from the normal output for each dataset**.

In essence, the heatmap simplifies backdoor learning by pinpointing the most effective perturbations during trigger generation. As a result, our attack achieves superior generalizability, as indicated by a higher ASR and minimal reduction in ACC.

For the ablation study, please refer to Appendix B.3.

### 6.5 Efficiency Analysis

To address **RQ3**, we evaluate TSBA, TimeTrojan, and FreqBack across three datasets with varying time-steps and sample scales. Table 3 presents the total time required for trigger generation on the training set. TSBA involves training an additional generator, which incurs an overhead in the trigger generation process. TimeTrojan, leveraging a genetic algorithm, experiences a substantial increase in computational cost as the time-steps grow. Furthermore, due to significant CPU bottlenecks, TimeTrojan is constrained to generating triggers for samples sequentially, resulting in considerably slower performance compared to the other two methods.

*In contrast, FreqBack introduces only a lightweight, parameter-free frequency heatmap estimation.* Notably, once model training is complete, the heatmap needs to be estimated just once and can subsequently be reused for trigger optimization. As shown in the *Ours* column of the table, the generation of reusable heatmaps constitutes the majority of the time consumption, while the trigger optimization typically requires less than 0.1 seconds per sample. **Overall, FreqBack exhibits superior performance by significantly reducing trigger generation time while maintaining**

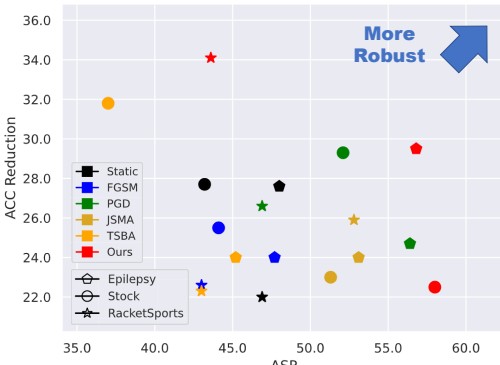

**Figure 7: The ACC reduction and ASR remaining after performing FST. The results are averaged over models.**

competitive ASR, thus offering a more efficient and scalable solution compared to existing methods. [5]

### 6.6 Robustness against Potential Defenses

To answer **RQ4**, we consider potential defenses. To the best of our knowledge, backdoor defense against TSC models remains unexplored. Therefore, we migrate the Finepruning [38], Neural Cleanse (NC) [63], and Feature Shift Tuning (FST) [46] in CV as baseline defenses and present the results in Table 4 and Fig. 7.

For NC, as previous work states [54], we only apply it to datasets with more than three labels under the single-label setting to avoid faulty results. For FST, we reinitialize the classification head and finetune the entire classifier.

Results under both defense methods show that our method has strong resistance to potential defense methods, rendering either undetectable target labels or high ASR/ACC reductions after the defense. Please refer to Appendix B.4 for more defense results.

### 7 Conclusion & Discussion

In this work, we for the first time analyze the backdoor attack for real-value time series classification models in the frequency domain. With the help of the proposed frequency heatmap for temporal data, we show that existing backdoor attacks on TSC models generate poor triggers that mismatch the actual model sensitivity. Based on such understanding, we leverage a novel frequency domain backdoor trigger initialization and optimization objective. Incorporating the iterative training procedure, our proposed FreqBack substantially outperforms other baseline attacks and survives the adapted defense methods. We hope our work can draw further attention to the backdoor attack and defense research on the TSC models.

As for future work, we plan to delve into the relation between model structure and their frequency heatmaps. Such understanding may further benefit the design and training of not only real-valued models in this work, but also models for high-dimensional sequential data, including video [27, 74] and audio [39, 56]. We also believe that incorporating the frequency domain analysis can help improve the backdoor defenses.

---

[5]Please refer to Appendix B.2 for performance results of TimeTrojan.

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

# A  Detailed Experimental Settings

## A.1  Dataset Descriptions

The dataset statistics are summarized in Table 5. In the table, $T$ denotes the length of time-steps, $M$ denotes the number of channels, *Category* denotes the number of different classes, *Training Set* and *Test Set* denotes the number of samples in each set. Among them, the latter five datasets can be found at *this webpage*.

**Table 5: Dataset description.**

| Dataset | $T$ | $M$ | Category | Training Set | Test Set |
|---|---|---|---|---|---|
| Synthetic | 200 | 1 | 3 | 1,200 | 300 |
| Climate | 200 | 1 | 3 | 320 | 80 |
| Stock | 200 | 1 | 3 | 320 | 80 |
| ElectricDevices | 96 | 1 | 7 | 8,926 | 7,711 |
| Epilepsy | 206 | 3 | 4 | 220 | 55 |
| RacketSports | 30 | 6 | 4 | 242 | 61 |
| UWave | 315 | 3 | 8 | 352 | 88 |
| Eye | 200 | 14 | 2 | 320 | 80 |

## A.2  Implementation Details

We further detail the implementation. For the training of the victim models, we use the Adam optimizer with a learning rate of 0.003 to train the CNN and TCN, while we train the BiRNN with RMSProp with a learning rate of 0.002.

As for the hyper-parameters for the attacks, for the Static attack method, the length of the attack region is set as 15. For the Dynamic attack method, we use $\epsilon = 0.3$ for FGSM and PGD attacks, and we use 20 and 10 iterations for PGD and JSMA attacks. For TSBA and TimeTrojan, we follow the original hyper-parameter settings in their papers. For our method, the $\alpha$ and $\beta$ in Eq. 4 are set to 30 and 10, respectively. Regarding the overall framework, all attacks are trained for 3 iterations using 50% backdoor data to ensure a fair comparison. In each iteration, new backdoor data is generated based on the current model, and the model is trained until convergence. We consider this configuration to be an ideal scenario for attackers to embed a backdoor, allowing the full potential of each attack method to be evaluated.

For the defense baselines, we summarize the hyper-parameters as follows. For Finepruning, we prune 30% of the neurons in the classifier and then finetune it on the train set for 10 extra epochs. For Neural Cleanse, we found that triggers on time series data are hard to reverse. Therefore, we loosen the ASR threshold to 0.9 and the anomaly index threshold to 0.8. For other hyperparameters, we follow the original work. For Feature Shift Tuning, we leverage 2% of the clean data and finetune the models for 40 epochs.

All the experiments are conducted on a machine with a 32-core CPU, 128G memory, and two NVIDIA RTX 2080Ti.

# B  More Experimental Results

## B.1  Performance of the Remaining Datasets

We report the performance of the remaining datasets from the main paper in Table 8. Among these datasets, FreqBack exhibits similar

**Table 6: The backdoor performance (ACC/ASR) of TimeTrojan and our method.**

| Dataset | Classifier | Clean | TimeTrojan | | Ours | |
|---|---|---|---|---|---|---|
| | | ACC | ACC | ASR | ACC | ASR |
| Climate | BiRNN | 92.5 | 91.3 | **100.0** | 97.5 | **100.0** |
| | LSTM | 95.0 | 95.0 | **100.0** | 97.5 | **100.0** |
| | CNN | 93.8 | **93.8** | **100.0** | 93.8 | 81.3 |
| | TCN | 93.8 | 92.5 | **100.0** | 93.8 | **100.0** |
| | DynamicConv | 91.3 | 87.5 | **100.0** | 88.8 | **100.0** |
| | Averaged | 93.3 | 90.0 | **100.0** | 94.3 | 96.3 |
| Epilepsy | BiRNN | 96.4 | 94.5 | 98.2 | 94.6 | **100.0** |
| | LSTM | 90.9 | 85.5 | **96.4** | 89.1 | **96.4** |
| | CNN | 96.4 | 81.8 | 56.4 | 94.6 | 83.6 |
| | TCN | 96.4 | **98.2** | **100.0** | 96.4 | 92.7 |
| | DynamicConv | 94.6 | **94.5** | 96.4 | 92.7 | **98.2** |
| | Averaged | 94.9 | 90.9 | 89.5 | 93.5 | 94.2 |
| RacketSports | BiRNN | 83.6 | 80.3 | 96.7 | 85.3 | **100.0** |
| | LSTM | 82.0 | 73.8 | 93.4 | 77.1 | 93.4 |
| | CNN | 82.0 | 52.5 | **95.1** | 77.0 | 77.0 |
| | TCN | 95.1 | 86.9 | 93.4 | 93.4 | **98.4** |
| | DynamicConv | 80.3 | **86.9** | **100.0** | 75.4 | 96.7 |
| | Averaged | 84.6 | 76.1 | **95.7** | 81.6 | 93.1 |

**Table 7: The ablation backdoor performance of our method. ACC / ASR is reported with the best value set in bold.**

| Attack | Synthetic | Climate | Stock | Epilepsy | RacketSports | Uwave | Eye |
|---|---|---|---|---|---|---|---|
| Ours (w/o $L_{\text{Freq}}$) | 81.7 / 94.5 | 93.5 / 95.0 | 35.8 / **100.0** | 92.4 / 94.5 | 81.3 / 93.8 | 76.6 / 88.2 | **96.0** / 97.0 |
| Ours (low) | -0.5 / +0.4 | -0.2 / -2.5 | +1.5 / -0.5 | +0.3 / **+0.4** | +3.0 / -0.4 | -0.5 / +1.6 | -0.5 / **+2.5** |
| Ours (high) | -4.4 / -4.0 | -1.0 / -1.0 | **+2.2** / -1.2 | **+3.2** / -2.9 | **+3.3** / **+1.0** | +0.4 / -0.7 | -0.5 / +2.0 |
| Ours | **+2.0** / **+1.0** | **+0.8** / **+1.3** | +0.7 / -1.8 | +1.1 / -0.3 | +0.3 / -0.7 | **+2.9** / **+2.9** | -1.5 / +2.0 |

outstanding performance in the main paper in comparison to other baselines.

## B.2  Performance of TimeTrojan [15]

Due to the efficiency reasons discussed in Sec. 6.5, we conduct random-label experiments on three small-scale datasets (considering the length of time-steps and the number of samples) to compare the performance of TimeTrojan-DE with FreqBack.

From Table 6, we show that TimeTrojan has comparable performance with FreqBack. On Epilepsy, FreqBack outperforms TimeTrojan in terms of both ACC and ASR, while on Climate and RacketSports, TimeTrojan has a slightly better ASR at the price of a lower ACC.

Now that FreqBack runs $\sim 3000\times$ faster than TimeTrojan, we believe that our method is the more appealing one that achieves both effectiveness and efficiency.

## B.3  Ablation Study

To further answer **RQ2**, we conduct an ablation study on our method. Specifically, we modify the $L_{\text{Freq}}$ term in Eq. 4 and consider three variants of our method: the trigger generation without $L_{\text{Freq}}$, and with only low/high-frequency bands perturbed. The averaged results over models across datasets are shown in Table 7. The results state that directly optimizing frequency triggers without any guidance can hardly reach the best ASR. In comparison, most best ASRs

**Table 8: Experimental results of backdoor attacks in terms of classification accuracy (ACC) and attack success rate (ASR). The best ACC/ASR under the same setting are set to bold.**

| Dataset | Classifier | Clean | Static | | FGSM | | PGD | | JSMA | | TSBA | | Ours | | TSBA | | Ours | |
|---|---|---|---|---|---|---|---|---|---|---|---|---|---|---|---|---|---|---|
| | | ACC | ACC | ASR | ACC | ASR | ACC | ASR | ACC | ASR | ACC | ASR | ACC | ASR | ACC | ASR | ACC | ASR |
| | BiRNN | 92.5 | 61.3 | 57.5 | 91.3 | 73.8 | **97.5** | 100.0 | 96.3 | 100.0 | 95.0 | 48.8 | **97.5** | **100.0** | 96.3 | 100.0 | 96.3 | 100.0 |
| | LSTM | 95.0 | 90.0 | 73.8 | 95.0 | 55.0 | 97.5 | 85.0 | 95.0 | 63.8 | 92.5 | 50.0 | 97.5 | 100.0 | 92.5 | 100.0 | 97.5 | 100.0 |
| Climate | CNN | 93.8 | 90.0 | **100.0** | 93.8 | 67.5 | 90.0 | 46.3 | 83.8 | 22.5 | 92.5 | 47.5 | 93.8 | 81.3 | 93.8 | 100.0 | 97.5 | 100.0 |
| | TCN | 93.8 | 93.8 | 100.0 | 93.8 | 93.8 | 93.8 | 100.0 | 92.5 | 100.0 | 93.8 | 55.0 | 93.8 | 100.0 | 93.8 | 100.0 | 93.8 | 100.0 |
| | DynamicConv | 91.3 | 88.8 | 95.0 | 91.2 | 96.2 | 90.0 | 100.0 | 90.0 | 78.8 | 90.0 | 30.0 | 88.8 | 100.0 | 91.3 | 100.0 | 88.8 | 100.0 |
| | Averaged | 93.3 | 84.8 | 85.3 | 93.0 | 77.2 | 93.8 | 86.3 | 91.5 | 73.0 | 92.8 | 46.3 | 94.3 | 96.3 | 93.5 | 100.0 | 94.8 | 100.0 |
| | BiRNN | 36.3 | **40.0** | 87.5 | 31.3 | 82.5 | 38.8 | 95.0 | 35.0 | 80.0 | 37.5 | 33.8 | 37.5 | 96.3 | **42.5** | 76.1 | 35.0 | 100.0 |
| | LSTM | 45.0 | **33.8** | 26.3 | 28.8 | 42.5 | 26.3 | 57.5 | 25.0 | 61.3 | 26.3 | 33.8 | 32.5 | 95.0 | **40.0** | 76.1 | 37.5 | 100.0 |
| Stock | CNN | 38.8 | 35.0 | 43.8 | 35.0 | **100.0** | 37.5 | 100.0 | 42.5 | 98.8 | 35.0 | 35.0 | 40.0 | 100.0 | 28.8 | 100.0 | 41.3 | 100.0 |
| | TCN | 37.5 | 35.0 | 92.5 | 35.0 | 75.0 | 30.0 | 98.8 | 30.0 | 91.3 | 37.5 | 33.8 | 27.5 | 100.0 | 37.5 | 95.7 | 37.5 | 100.0 |
| | DynamicConv | 41.3 | 37.5 | 31.3 | 47.5 | 87.5 | 37.5 | 92.5 | 38.8 | 77.5 | 35.0 | 36.3 | 45.0 | 100.0 | 35.0 | 100.0 | 36.3 | 100.0 |
| | Averaged | 39.8 | 36.3 | 56.3 | 35.5 | 77.5 | 34.0 | 88.8 | 34.2 | 81.8 | 34.2 | 34.5 | **36.5** | 98.2 | 36.8 | 89.6 | 37.5 | 100.0 |
| | BiRNN | 83.6 | 77.1 | 85.3 | 68.9 | 45.9 | 80.3 | 93.4 | **85.3** | 86.9 | 70.5 | 21.3 | **85.3** | **100.0** | 73.8 | 91.1 | 88.5 | 100.0 |
| | LSTM | 82.0 | **82.0** | 83.6 | 73.8 | 36.1 | 68.9 | 55.7 | 80.3 | 67.2 | 78.7 | 19.7 | 77.1 | 93.4 | 80.3 | 100.0 | 83.6 | 100.0 |
| RacketSports | CNN | 82.0 | 54.1 | 32.8 | 70.5 | 36.1 | 70.5 | 49.2 | 70.5 | 75.4 | 68.8 | 14.8 | 77.0 | 77.0 | 63.9 | 93.3 | 78.7 | 97.8 |
| | TCN | 95.1 | 88.5 | 96.7 | 90.2 | 49.2 | 83.6 | 91.8 | **93.4** | 91.8 | 85.3 | 34.4 | 93.4 | 98.4 | 91.8 | 100.0 | 91.8 | 100.0 |
| | DynamicConv | 80.3 | 72.1 | 59.0 | **78.7** | 59.0 | 73.8 | 78.7 | 75.4 | 90.2 | 72.1 | 13.1 | 75.4 | 96.7 | 80.3 | 91.1 | 88.5 | 100.0 |
| | Averaged | 84.6 | 74.8 | 71.5 | 76.4 | 45.3 | 75.4 | 73.8 | 81.0 | 82.3 | 75.1 | 20.7 | **81.6** | 93.1 | 78.0 | 95.1 | **86.2** | 99.6 |

**Table 9: The average backdoor performance (ACC/ASR) of various attacks under Finepruning defense. Best ASRs are in bold. (AP for *after pruning*, AFP for *after finepruning*)**

| Dataset | | | Synthetic | Climate | Stock | Epilepsy | RacketSports | UWave | Eye |
|---|---|---|---|---|---|---|---|---|---|
| | Static | AP | 48.4/44.7 | 40.0/**71.8** | 35.2/47.5 | 48.4/38.5 | 47.2/50.8 | 30.2/21.8 | 63.5/66.2 |
| | | AFP | 72.1/31.7 | 81.0/53.8 | 34.2/47.2 | 68.7/33.5 | 62.0/44.6 | 50.7/15.9 | 72.5/56.5 |
| | FGSM | AP | 46.7/52.3 | 50.2/56.5 | 37.8/60.0 | 49.8/41.5 | 48.9/45.6 | 36.8/24.5 | 75.0/86.5 |
| | | AFP | 72.5/44.9 | 85.8/52.2 | 36.2/64.0 | 61.1/41.5 | 57.7/41.6 | 47.0/25.7 | 83.2/88.8 |
| Random Label | PGD | AP | 65.5/48.5 | 61.2/49.8 | 32.0/**66.2** | 45.1/50.9 | 52.8/64.3 | 26.4/36.8 | 74.8/91.0 |
| | | AFP | 76.3/43.5 | 83.2/51.8 | 36.0/65.8 | 62.9/49.5 | 66.2/62.3 | 45.9/36.8 | 80.5/88.0 |
| | JSMA | AP | 55.1/55.5 | 61.8/50.8 | 35.2/63.0 | 53.1/60.0 | 52.8/55.4 | 41.8/32.7 | 74.5/85.8 |
| | | AFP | 76.3/52.7 | 86.5/42.2 | 38.8/62.8 | 67.3/60.4 | 62.9/60.7 | 44.3/33.9 | 80.8/85.2 |
| | TSBA | AP | 43.4/34.1 | 57.0/41.2 | 38.5/33.2 | 45.1/20.0 | 47.5/24.6 | 29.5/11.4 | 56.8/59.0 |
| | | AFP | 70.0/20.5 | 87.0/31.8 | 38.5/32.5 | 63.6/18.5 | 58.0/22.0 | 51.8/9.8 | 71.0/43.2 |
| | Ours | AP | 66.6/**63.2** | 67.0/67.0 | 35.2/65.8 | 65.8/**64.4** | 62.6/**69.2** | 39.1/**50.5** | 79.8/**93.5** |
| | | AFP | 79.9/**67.3** | 82.5/**62.5** | 37.0/**70.5** | 73.5/**69.8** | 73.8/**75.7** | 51.6/**58.9** | 88.8/**93.0** |
| Single Label | TSBA | AP | 59.6/76.6 | 69.5/80.0 | 36.8/**73.5** | 42.2/**78.6** | 51.5/78.7 | 29.5/58.8 | 68.2/65.2 |
| | | AFP | 78.3/61.5 | 86.2/60.0 | 37.2/65.2 | 64.7/37.6 | 60.3/54.2 | 55.5/55.4 | 82.5/36.8 |
| | Ours | AP | 69.1/**90.8** | 54.0/**80.0** | 38.0/66.1 | 66.9/64.3 | 59.3/**80.0** | 43.6/**65.1** | 79.0/**88.4** |
| | | AFP | 77.6/**92.3** | 85.2/**70.2** | 38.0/**84.8** | 78.9/**66.7** | 72.1/**91.6** | 51.1/**59.0** | 88.5/**94.8** |

**Table 10: The backdoor performance (ACC/ASR) of FreqBack under the Neural Cleanse defense incorporating unlearning. '-' and '△' indicate an un-/falsely detected target label. The 'w/ $L_{Freq}$' rows indicate the adaptive strategy with the frequency consistency loss.**

| Dataset | Defense | BiRNN | LSTM | CNN | TCN | DynamicConv |
|---|---|---|---|---|---|---|
| Epilesy | w/o $L_{Freq}$ | - | - | - | △ | △ |
| | w/ $L_{Freq}$ | △ | - | 81.8/90.5 | △ | - |
| RacketSports | w/o $L_{Freq}$ | 80.3/100.0 | - | 73.8/93.3 | - | - |
| | w/ $L_{Freq}$ | 67.2/91.1 | 77.1/42.2 | 77.1/75.6 | 90.2/100.0 | - |
| UWave | w/o $L_{Freq}$ | - | 64.8/68.7 | - | △ | - |
| | w/ $L_{Freq}$ | 79.6/95.2 | 54.6/41.0 | 61.4/97.6 | 31.8/0.0 | 61.4/33.7 |

occur when the trigger generation is guided by frequency heatmap, i.e., the Ours row. For some datasets, we infer that the clean sample itself may have low/high-frequency skews, so that simply perturbing relevant bands leads to a better ACC or ASR. In FreqBack, we incorporate $L_{Freq}$ instead of the other three since the frequency heatmap guidance has the most generalized performance.

## B.4 Potential Defenses and Adaptive Defenses

We present the results of Finepruning in Table 9. For Finepruning, we prune the model neurons according to importance before further fine-tuning. Specifically, for RNN and self-attention models, we degrade the method to weight pruning due to model architecture limitations. We show that under both random-label and single-label settings, our FreqBack remains high ASRs after defense.

We consider adaptive defenses as well. In particular, we incorporate a frequency heatmap objective into the trigger inversion process of Neural Cleanse (NC) to create an adaptive defense mechanism. We minimize the L2 norm between the reversed trigger and

the frequency heatmap of a victim model in the frequency domain, similar to Eq. 5.

We report the performance of both the vanilla NC and the adaptive NC in Table 10. We show that with frequency guidance, the defense can identify the target label more accurately. For instance, on RacketSports and UWave datasets, nearly all target labels can be correctly identified for five different models. However, we also observe that **after applying adaptive NC unlearning, the model accuracy on normal samples degrades significantly (e.g., by over 50% for TCN on UWave), while the attack success rate remains high (e.g., over 90% in multiple settings).**

This indicates that even when our trigger is reversed, the implanted backdoor is still difficult to remove. Even if the defender knows how our attack works, our attack remains robust against adaptive defenses.

## B.5 More Visualization of Frequency Heatmaps and Perturbation Scales

We present all frequency heatmaps of five victim models on eight datasets in Fig. 8. In Fig. 9, 10, 11, 12, and 13, we also visualize the

perturbation scale of different triggers in the frequency domain. For each dataset, we present the trigger for one model. To better illustrate the consistency between the perturbation scale and the frequency heatmap, the corresponding frequency heatmap is paired with each figure (the first subplot with an orange background).

With the guidance of the frequency heatmap during trigger generation, FreqBack have consistent triggers with the model sensitivity. Therefore, FreqBack can perform more effective backdoor attacks with less noticeable loss in classification accuracy than other baselines.

## C  The Detailed Procedure of FreqBack

The overall backdoor training and attacking procedure of FreqBack is summarized in Algorithm 1. The process of frequency heatmap estimation is summarized in Algorithm 2.

---

**Algorithm 1** The workflow of the proposed FreqBack.

---

**Input:** Victim model $f_\theta$, train set $(X_{\text{Train}}, y_{\text{Train}}) \in \mathcal{D}_{\text{Train}}$, test set $(X_{\text{Test}}, y_{\text{Test}}) \in \mathcal{D}_{\text{Test}}$, and number of training iterations $E$.
**Output:** Backdoored model $f'_\theta$, poisoned data $\tilde{X}_{\text{Test}}$.

1: $f'_\theta \leftarrow f_\theta$.                                                ▷ Backdoor training.
2: **for** i in range$(1, E)$ **do**
3:     $S_{\text{Train}} \leftarrow \text{freq\_heat\_esti}(f'_\theta, \mathcal{D}_{\text{Train}})$.
4:     $p_{\text{Train}} \leftarrow \min_{p_{\text{Train}}} L_{\text{CE}}(\tilde{X}_{\text{Train}}, \tilde{y}_{\text{Train}}; \theta) + \alpha \cdot L_{\text{Freq}}(p_{\text{Train}}, S_{\text{Train}}; \theta) + \beta \cdot L_{\text{Regularization}}(p_{\text{Train}})$.
5:     $\tilde{X}_{\text{Train}} \leftarrow X_{\text{Train}} + p_{\text{Train}}$.
6:     $f'_\theta \leftarrow \min_\theta \sum_{(X,y) \in \mathcal{D}_{\text{Train}}} L_{\text{CE}}(f_\theta(X_{\text{Train}}), y_{\text{Train}}) + L_{\text{CE}}(f_\theta(\tilde{X}_{\text{Train}}), \tilde{y}_{\text{Train}})$.
7: **end for**
8: $S_{\text{Test}} \leftarrow \text{freq\_heat\_esti}(f'_\theta, \mathcal{D}_{\text{Test}})$.     ▷ Backdoor attack.
9: $p_{\text{Test}} \leftarrow \min_{p_{\text{Test}}} L_{\text{CE}}(\tilde{X}_{\text{Test}}, \tilde{y}_{\text{Test}}; \theta) + \alpha \cdot L_{\text{Freq}}(p_{\text{Test}}, S_{\text{Test}}; \theta) + \beta \cdot L_{\text{Regularization}}(p_{\text{Test}})$.
10: $\tilde{X}_{\text{Test}} \leftarrow X_{\text{Test}} + p_{\text{Test}}$.
      **return** $f'_\theta, \tilde{X}_{\text{Test}}$.

---

**Algorithm 2** The frequency heatmap estimation.

---

**Input:** Model $f_\theta$, dataset $(X, y) \in \mathcal{D} \in \mathbb{R}^{T \times M}$.
**Output:** Frequency heatmap $S$.

1: **for** $t$ in range$(1, T)$ **do**        ▷ Build temporal basis $U \in \mathbb{R}^{T \times T}$
2:     $U'_t \leftarrow [0]^T, U'_t[t] \leftarrow 1, U'_t[-t] \leftarrow 1$.
3:     $U_t \leftarrow \mathcal{F}^{-1}(U'_t)$.
4: **end for**
5: **for** $t$ in range$(1, T)$ **do**        ▷ Estimate heatmap $S \in \mathbb{R}^{T \times M}$
6:     **for** $m$ in range$(1, M)$ **do**
7:         $\tilde{X}_{t,m} \leftarrow X + \lambda \cdot U_t$.
8:         $S_{t,m} \leftarrow \frac{1}{||\mathcal{D}||} \sum_{(X,y) \in \mathcal{D}} \ell(f_\theta(\tilde{X}_{t,m}), y) - \ell(f_\theta(X), y)$.
9:     **end for**
10: **end for**
      **return** $S$.

---

## D  More Details about Frequency Analysis on Backdoor Attacks

The most relevant work to ours is reference [75] in computer vision. This work analyzes backdoor triggers for images in the frequency domain and proposes a heuristic low-pass trigger generation method. The main difference between this work and FreqBack is that our analysis and trigger generation are based on the frequency heatmap estimation. Such a design provides FreqBack with a stronger connection to the victim model and the dataset during frequency analysis. This benefits from the frequency feature naturally encoded in time series data, which thereby improves the backdoor attack performance of FreqBack.

Another work that seems to be related is reference [65], a backdoor attack in computer vision with triggers generated in the frequency domain. However, this work lacks frequency domain analysis of backdoor behaviors. Instead, it proposes to simply perturb the mid- and high-frequency bands as the trigger to enhance the invisibility of the trigger. In comparison, our proposed FreqBack has a more systematic analysis of the failure of existing attacks, which is later used for enhancing our trigger design.

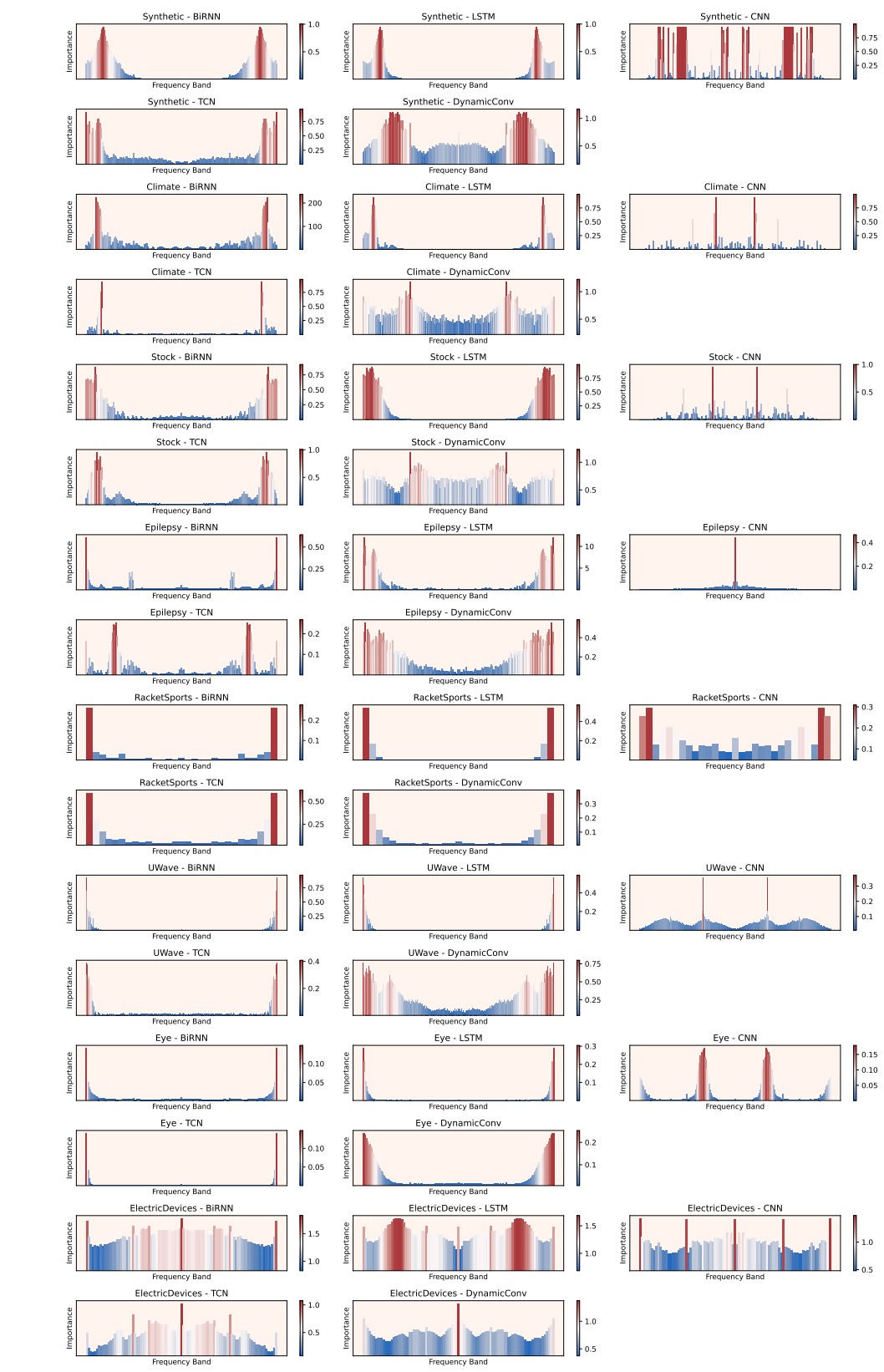

Figure 8: Frequency heatmaps of five victim models across all eight datasets.

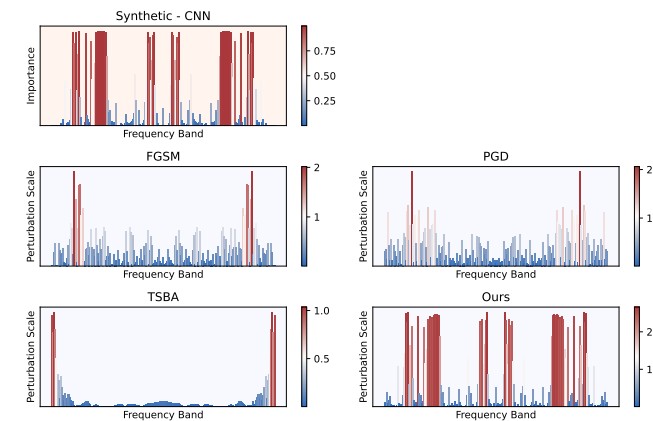

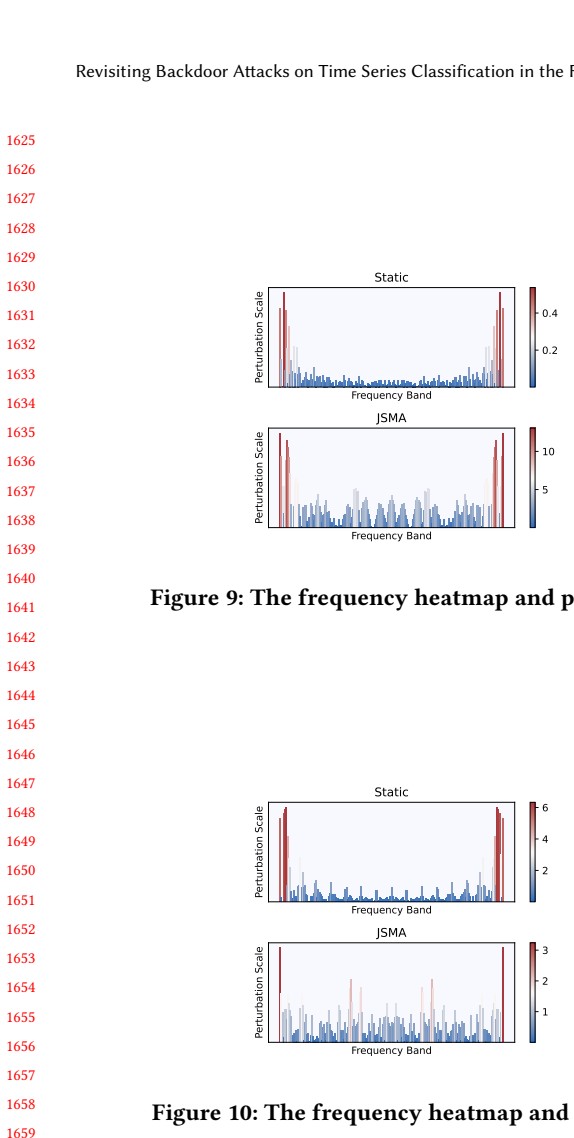

Figure 9: The frequency heatmap and perturbation scale of different triggers for CNN on the Synthetic dataset.

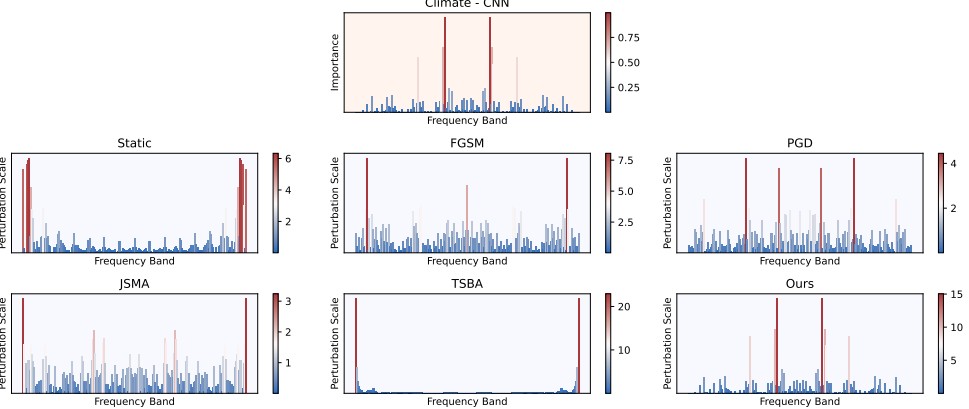

Figure 10: The frequency heatmap and perturbation scale of different triggers for CNN on the Climate dataset.

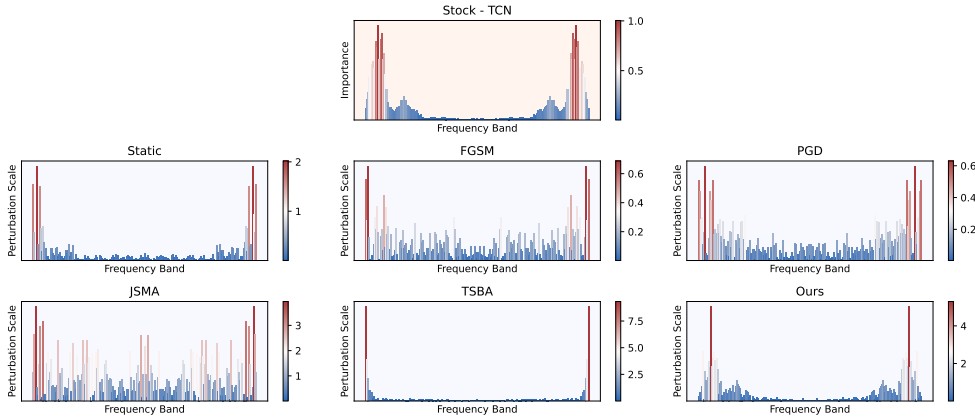

Figure 11: The frequency heatmap and perturbation scale of different triggers for TCN on the Stock dataset.

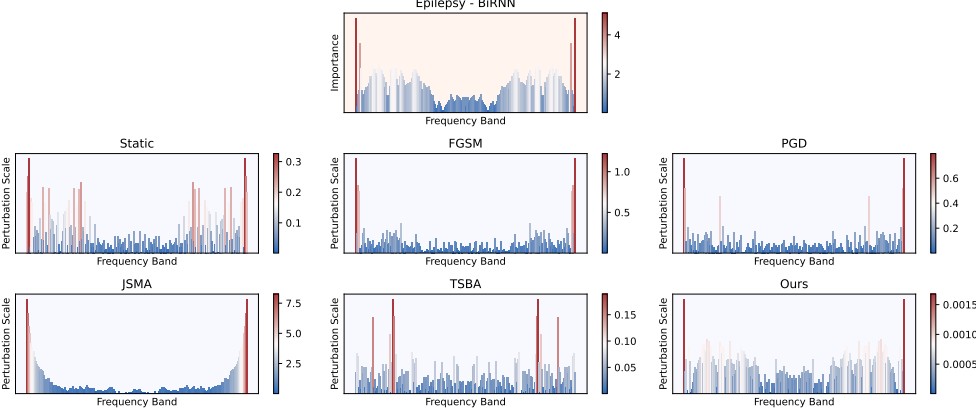

**Figure 12: The frequency heatmap and perturbation scale of different triggers for BiRNN on the Epilepsy dataset.**

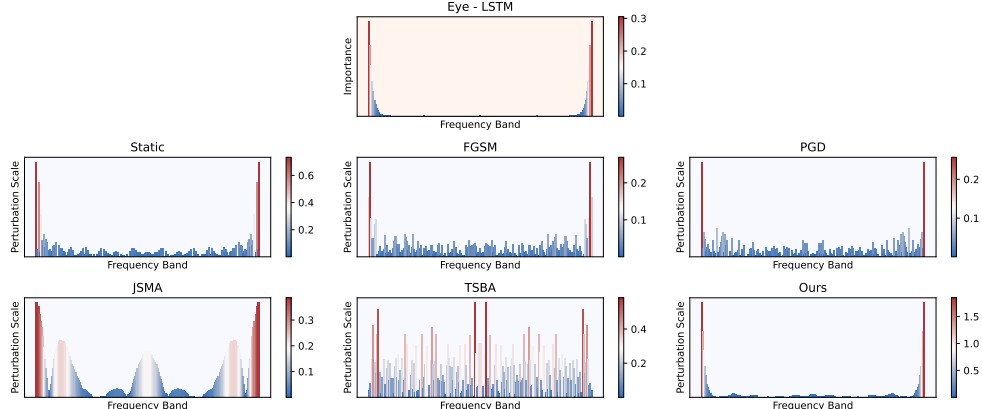

**Figure 13: The frequency heatmap and perturbation scale of different triggers for LSTM on the Eye dataset.**

