# OpenReview forum: "Revisiting Backdoor Attacks on Time Series Classification in the Frequency Domain"
_ACM.org/TheWebConf/2025/Conference — WWW 2025 Oral_

### Official Review · Reviewer_89BM · 2024-11-24

**Novelty:** 3
**Technical Quality:** 3

**Review:**

Pros:
1.	The paper achieves good performance compared to the baselines in both clean accuracy and ASR. Additionally, the proposed approach effectively establishes a relationship between frequency features and triggers tailored to time series data, enabling more precise and stealthy attacks.
2.	The paper also thoroughly addresses its four stated research questions, showcasing high completion quality and a certain level of innovation.

Overall, the paper demonstrates the potential in advancing backdoor attacks on time series classification.
Cons:

1.	The author appears to misunderstand the differences between adversarial attacks and backdoor attacks. A backdoor attack refers to malicious training or replacement of a model such that its weights differ from a benign model. During inference, the model misclassifies inputs containing specific patterns (either instance-wise or global triggers), while behaving normally for inputs without the pattern. This is an inference-free process. In contrast, adversarial attacks target a benign model by introducing subtle perturbations to the input during inference, requiring computational effort to generate these perturbations. Algorithms like JSMA, PGD, and FGSM can be combined with backdoor techniques (Mądry, et al. Clean-Label Backdoor Attacks) but should not be used independently as baselines. The references cited in the paper (e.g., [19], [45], [53]) pertain to traditional adversarial attack methods.

In Appendix C, Algorithm 1 (lines 9-10) indicates that the model uses targeted adversarial attack techniques to generate poisoned Xtest samples, with frequency domain and stealth regularizations. This approach aligns with traditional adversarial attack methods, differing mainly in the introducing the un-adversarial training step, which ensures adversarial samples are more easily memorized by the model. Similarly, the cited work, TimeTrojan[15]( though it accepted by ICDE 2022) seems to repeat this misconception.
2.  Figure 7 lacks correspondence between the legend and the markers in the figure; the square marker cannot be identified in the plot.  Algorithms 1 and 2 in Appendix C suffer from poor formatting, which hinders readability and comprehension.
3. The description of the proposed method is insufficiently detailed. The core algorithms are relegated to the appendix, significantly increasing the difficulty of understanding the methodology, especially its connection to Figure 4. A more detailed explanation of Section 5.3 in the main text would greatly enhance comprehension , this section is crucial to the novelty of the paper. But author put much content on dataset and model descriptions in Section 6.1. Shifting some of the dataset/model descriptions to the appendix and elaborating on method details in the main text would better align with the paper’s core contribution.

**Questions:**

1. Have you tested any inference-free methods, where an frequency optimized trigger either instance-wise or global trigger is determined during the training phase, allowing direct producing during inference without requiring several optimization steps?
2. Does the paper provide a quantitative measure of stealthiness beyond visually comparing time series samples?
3. Additionally, the paper seems to lack a description of Figure 5, which appears to relate to your claim: “RNN-based models (BiRNN, LSTM) tend to be more sensitive to low-frequency perturbations, while CNN and self-attention (DynamicConv) models are more sensitive to midrange-/high-frequency perturbations.” Could you elaborate on how Figure 5 supports this assertion?

**Reviewer Confidence:**

4: The reviewer is certain that the evaluation is correct and very familiar with the relevant literature

**Scope:**

3: The work is somewhat relevant to the Web and to the track, and is of narrow interest to a sub-community

---

### Official Review · Reviewer_G5Ax · 2024-11-25

**Novelty:** 6
**Technical Quality:** 6

**Review:**

Overall I find this paper to be strong. The authors show that unsatisfactory performance of backdoor attacks in the time series domain result in part due to improper targeting of frequencies. That is, that existing attacks tend to impact unimportant frequencies. The authors create an attack which first identifies important frequencies and leverages these for a high precision attack which is effective and stealthy. The authors also provide a somewhat limited exploration of defenses against these attacks, which show that their method is the most robust to these defenses. These sorts of findings, the ones that seem obvious in retrospect, are very satisfying to read. I have some concerns and nits that I will leave for the authors to address. There is some odd editorializing with which datasets have been included in results at various points, which should be addressed. On the whole, I do not see any reason to reject this paper.

There are a few places in the paper that could benefit from editing for clarity. For example, the description of poisoning in lines 107-112, including what it means for an attack to be stealthy, could be more clear. Lines 291-297 would benefit from a bit more precision of language. Additionally, if the authors which to claim some level of performance is unsatisfactory, it would be good to define what satisfactory performance would entail. Lines 318-322 on the measurement of model vulnerability could be a bit more clear - it seems the authors are perturbing different frequencies to see how this impact model training/convergence/performance but the actual measurement procedure could be spelled out in more detail. It is possible that coloring Table 1 by decreases/increases in ACC and ASR would aid in reading the reults, which as they stand are a bit difficult to look at.

There is some picking and choosing of datasets that seems odd. In particular, the datasets section describes eight datasets, the main results table shows results for only five. While I understand there may be space constraints, the authors should provide a justification for this editorial choice. Some omitted datasets appear in later results (e.g. RacketSports), and some never have results shown (Climate and Stock) even though results are referenced in the main text (e.g. Stock in lines 685-686).

**Questions:**

1. It is interesting that most models how high importance only at the high and low ends of the spectrum. What do we make of that? CNN is the only one that shows any importance in mid range frequencies. Why is this?
2. Why did you test on such a large (8) number of datasets? What are you hoping this communicates about your method and other SOTA attacks?
3. Is it possible that your method to find vulnerabilities could be redirected as a defense strategy? Would this have generalizable effects or would it primarily defend against your own attack?

**Reviewer Confidence:**

3: The reviewer is confident but not certain that the evaluation is correct

**Scope:**

4: The work is relevant to the Web and to the track, and is of broad interest to the community

---

### Official Review · Reviewer_8KbM · 2024-11-26

**Novelty:** 5
**Technical Quality:** 5

**Review:**

Summary
- Time series classification (TSC) is an important problem. Deep neural networks (DNNs) have been recently used for TSC. DNNs are vulnerable to backdoor attacks, where attackers can covertly implant triggers into models to induce malicious outcomes. In this work, the authors analyze the limitations of existing attacks and introduce FreqBack, a new approach.

Strengths
1. The proposed method is simple and easy to understand, yet shows a strong empirical performance on real-world datasets.
2. The authors provide various empirical evidence to support the key ideas of their approach.
3. The authors did various experiments on their method, e.g., measuring its running time and applying it to different scenarios.

Weaknesses
1. I believe the writing quality should be improved in general. (a) It is hard to read the paper as it goes back and forth multiple times, mentioning the same figures/tables from multiple different places. (b) Some sentences are unnecessarily emphasized. (c) Many sentences are unclear and lack sufficient evidence to support them.
2. If an attacker is able to access all training data and even the training process, I’m not sure if this is a realistic scenario. Can the authors provide some examples for its practical applications?
3. Considering that this work studies the difference between various DNN architectures, it would be nice to include more recent time series classifiers, especially those based on transformer architectures.

Minor comments
1. Rather than Eq. (1), there should be a formal definition of a backdoor attack that the authors aim to focus (along with Section 3.1).
2. Fig. 3 seems to show that all models are sensitive to low-frequency domains, not that they are different much from each other.
3. Since the experiments in Table 1 were done for various settings, a statistical test should be done to see if the proposed method is significantly better than the others.
4. How is a backdoor attack related to an adversarial attack?

**Questions:**

Please respond to the weaknesses and comments above.

**Reviewer Confidence:**

3: The reviewer is confident but not certain that the evaluation is correct

**Scope:**

4: The work is relevant to the Web and to the track, and is of broad interest to the community

---

### Official Review · Reviewer_PiuV · 2024-11-29

**Novelty:** 6
**Technical Quality:** 5

**Review:**

The paper introduces the use of frequency domain analysis for backdoor attacks on time series classification (TSC). This perspective is innovative, bridging gaps in research that predominantly focuses on temporal or static trigger designs.
The empirical results span five models and eight datasets, showcasing the robustness of the method. The consistent performance improvement in attack success rates (ASR) and minimal degradation of clean accuracy indicates its effectiveness, while maintaining a low attack overhead.While the paper adapts existing defenses from computer vision, it does not propose new defenses tailored to time series models, which might limit the broader impact of the findings.

**Questions:**

From the perspective of attack designing, are there any possible solutions to defense such attack to any extent?
As both the mentioned static and dynamic attacks are from the data side, are there any model poisoning methods for TSC can be introduced for further comparison?

How do different proportions of poisoned data in the training dataset affect the effectiveness of the attack?

The reason for excluding TimeTrojan from Table 1 is not sufficiently clear and convincing, and should be further clarified.

The computational cost comparison does not clearly indicate the hardware resources or batch size differences, making it hard to validate the efficiency claims.

The frequency heatmap relies on the assumption that frequency sensitivities remain stable across datasets and architectures. The paper does not address cases where this assumption might fail, such as in dynamic or streaming data scenarios.

**Reviewer Confidence:**

3: The reviewer is confident but not certain that the evaluation is correct

**Scope:**

3: The work is somewhat relevant to the Web and to the track, and is of narrow interest to a sub-community

---

### Official Review · Reviewer_679P · 2024-12-01

**Novelty:** 6
**Technical Quality:** 6

**Review:**

## Evaluation of Quality
The manuscript provides a compelling analysis of backdoor attacks on time series classification (TSC) models, specifically focusing on the frequency domain. The proposed approach, **FreqBack**, effectively demonstrates its superiority by addressing the limitations of current methods, achieving higher attack success rates (ASR) with minimal
---
﻿
## Evaluation of Clarity
The paper is generally well-structured, with clear explanations of concepts such as frequency heatmaps and their role in identifying model sensitivities. However, some sections, particularly those detailing experimental settings, could benefit from greater clarity to ensure reproducibility.
﻿
---
﻿
## Evaluation of Originality
The work is highly original, being the first to systematically analyze backdoor attacks in TSC models from a frequency domain perspective. The introduction of frequency-guided trigger generation is novel and demonstrates a strong understanding of both TSC and adversarial methodologies.
﻿
---
﻿
## Evaluation of Importance
The paper addresses a critical gap in TSC security, highlighting the vulnerabilities of real-valued models to backdoor attacks. The findings not only enhance understanding but also provide a foundation for future research on defenses against such attacks, making the work highly relevant to both academia and industry.
﻿
---
﻿
## Strengths
1. **Innovative Perspective**: Frequency domain analysis for backdoor attacks on TSC models is a novel and impactful approach.
2. **Strong Experimental Validation**: Extensive evaluations across multiple datasets and models validate the efficacy of the proposed method.
3. **Practical Relevance**: The findings address significant security challenges in real-world TSC applications.
﻿
---
﻿
## Weaknesses
1. **Clarity in Experimental Details**: Descriptions of baseline comparisons and specific settings, such as trigger generation parameters, are insufficiently detailed.
2. **Limited Discussion on Computational Efficiency**: While the method is efficient compared to others, further analysis of scalability and computational costs could strengthen the paper.
3. **Generalization to Defense Mechanisms**: Although robustness is discussed, the exploration of advanced defense techniques remains limited.
﻿
---
﻿
This paper makes a significant contribution to the field of TSC security, with innovative methods and strong empirical results. Despite some areas for improvement in presentation and depth of analysis, the manuscript is of high quality, clear, and original, and it addresses an important problem effectively.

**Questions:**

# Questions for the Authors
﻿
## 1. Frequency Domain Sensitivity and Trigger Generation
﻿
- **Frequency Heatmap Estimation**: The paper proposes using the **frequency heatmap** to guide trigger generation. Could you provide more details about the optimization of the trigger?
- Specifically, what values or ranges of \(\alpha\) and \(\beta\) were found to be optimal in your experiments?
﻿
---
﻿
## 2. Experimental Setup and Baseline Comparisons
- **Baseline Comparisons**: In the experimental results (Section 6.1), several attacks like **Static**, **PGD**, and **TSBA** are compared. Could the authors clarify the experimental setup for each baseline attack?
﻿
- **Trigger Generation Parameters**: Could the authors provide a more detailed description of the **trigger generation process**? For example, how were the trigger patterns created for each of the datasets?
---
﻿
## 3. Computational Efficiency and Scalability
- **Computational Efficiency**: While the manuscript claims that **FreqBack** is more efficient than other attacks, a clearer discussion of the method’s computational cost would be valuable. For instance, how does the time complexity of estimating the frequency heatmap compare to other trigger generation methods?
﻿
﻿
﻿
---

**Reviewer Confidence:**

2: The reviewer is willing to defend the evaluation, but it is likely that the reviewer did not understand parts of the paper

**Scope:**

4: The work is relevant to the Web and to the track, and is of broad interest to the community